# Altered Expression of *C4BPA* and *CXCL1* Genes in the Endometrium of Patients with Recurrent Implantation Failure after In Vitro Fertilization and Thin Endometrium

**DOI:** 10.3390/diagnostics14171967

**Published:** 2024-09-06

**Authors:** Gaukhar Kurmanova, Yeldar Ashirbekov, Almagul Kurmanova, Nagima Mamedaliyeva, Gaukhar Moshkalova, Gaini Anartayeva, Damilya Salimbayeva, Aidana Tulesheva

**Affiliations:** 1Medicine and Healthcare Faculty, Al Farabi Kazakh National University, 71 Al-Farabi Avenue, Almaty 050040, Kazakhstan; 2M. Aitkhozhin Institute of Molecular Biology and Biochemistry, 86 Dosmukhamedov Street, Almaty 050012, Kazakhstan; 3Scientific Center of Obstetrics, Gynecology and Perinatology, 125 Dostyk Ave., Almaty 050010, Kazakhstan

**Keywords:** endometrial receptivity, recurrent implantation failure, thin endometrium, transcriptomics

## Abstract

Currently, recurrent implantation failure (RIF) after in vitro fertilization is a problem that is commonly faced by reproductive specialists. The phenomenon of a thin endometrium in RIF patients is not yet completely understood or sufficiently treated. This study aimed to reveal the dysregulated expression of selected genes between RIF patients with a thin endometrium and fertile women. Endometrial samples were collected in the implantation window (21–24 days of the natural menstrual cycle) from RIF patients (*n* = 20) and fertile women (*n* = 14). Ten genes were chosen as target genes regarding their possible relations with the implantation process. The endometrial gene expression levels showed differences in RIF samples compared to fertile samples. Significant downregulation was observed for the *CXCL1* (*p* = 0.005) and *C4BPA* (*p* = 0.03) genes. There was no statistically significant difference between the RIF group and the fertile group in the expression of eight genes: *CXCL8*, *HPRT1*, *MMP10*, *INFG*, *VEGFB*, *HAND2*, *IL-15*, and *TNC* (*p* > 0.05). The use of a combination of two markers (*C4BPA + CXCL1*) allows for the good discrimination of RIF patients from fertile women (AUC 0.806).

## 1. Introduction

Recurrent implantation failure (RIF) after in vitro fertilization (IVF) and embryo transfer (ET) is a common problem faced by fertility specialists [1,2,3]. RIF is diagnosed when high-quality embryos fail to implant in at least three consecutive IVF attempts [2].

One of the critical factors for implantation failure is considered to be a thin endometrium, in which the thickness of the endometrium is less than 7 mm during the “implantation window”. The pathophysiological features of a “thin endometrium” include insufficient glandular epithelial growth, decreased growth factor expression, and the depletion of blood vessels [4]. In a thin endometrium in the proliferative phase, the local microenvironment is characterized by a decrease in stromal and epithelial cells, natural killer cells, and T cells and an increase in cellular senescence in perivascular cells [5]. The thin endometrium observed in patients with RIF remains a challenge in terms of understanding and effectively treating these conditions.

Studies over the last decade have shed light on structural changes in the receptive endometrium, which include the appearance of pinopodes, changes in the expression of endometrial receptors, and the secretion of regulatory factors [6,7]. All of these changes are associated with the transcriptional activity of the genes that control these processes [8]. The transcriptomic analysis of the endometrium has made it possible to begin to decipher the mechanisms of the implantation process and identify potential molecular biomarkers of endometrial receptivity [9,10].

Current trends in genomics and advances in bioinformatics allow the simultaneous analysis of up- and downregulated genes involved in the phenomenon of endometrial receptivity [11]. A significant number of genes are differentially expressed between the pre-receptive and receptive endometrial phases in healthy women, and some genes are also differentially expressed in women with implantation failure and in patients undergoing infertility treatment [12,13,14,15]. It is difficult to identify a limited number of genes that predict endometrial receptivity [16]. Therefore, the genes identified in several studies may become potential biomarkers of endometrial receptivity [17].

Although various mechanisms have been proposed to explain RIF, most studies investigating the meta-signatures of genes associated with endometrial receptivity disorders point to the importance of the abnormal activation of the innate immune system [18,19,20]. A gene ontology analysis showed that the most widely represented endometrial receptivity genes were the groups responsible for vascular proliferation [21] and immunological activity [22,23].

Since embryo implantation is an immunologically based process, the immunological profiling of endometrial receptivity is the most promising strategy in terms of predicting endometrial receptivity [11]. Therefore, in this study, we selected genes responsible for immune pathways and showing altered expression in the receptive phase in healthy women. We focused on genes responsible for the pro-inflammatory signaling cascade, i.e., *C-X-C motif chemokine ligand 8 (CXCL8)* and *C-X-C motif chemokine ligand 1 (CXCL1*); the complement cascade, i.e., *Complement Component 4 Binding Protein Alpha* (*C4BPA*); the breakdown of the extracellular matrix in normal physiological processes, such as embryonic development, reproduction, and tissue remodeling, i.e., *matrix metallopeptidase 10 (MMP10*); the generation of purine nucleotides, i.e., *hypoxanthine phosphoribosyltransferase 1 (HPRT1*); cell adhesion, i.e., *Tenascin C (TNC*); the vascular proliferation pathway, i.e., *vascular endothelial growth factor B* (*VEGF-B)*; and the innate and adaptive immune systems, i.e., *interferon gamma (INFG*), *heart- and neural crest derivative-expressed transcript 2 (HAND2*) and *interleukin 15 (IL-15*). *HAND2* regulates *IL-15*, which is required for the activation and survival of uterine natural killer cells [22,23]. Another reason for the failure of implantation is a decrease in the activity of the *Tenascin C (TNC*) gene, which is involved in reducing cell adhesion on the surface of the endometrium, facilitating the attachment of the embryo to the endometrium. In previous studies, the *TNC* gene expression level was significantly (*p* < 0.05) downregulated in RIF samples compared to the control group [19,24].

Thus, 10 immune response genes were selected (*CXCL8*, *CXCL1*, *HPRT1*, *MMP10*, *INFG*, *C4BPA*, *TNC*, *VEGFB*, *HAND2*, and *IL-15*), the activity of which was increased during the receptive endometrial phase in healthy women. In addition, two of them (*C4BPA* and *TNC*) showed downregulation upon implantation failure in women. It is assumed that a decrease in the expression of these transcriptomes indicates implantation failure in assisted pregnancies with a thin endometrium. The research question posed in this study was whether the selected genes would show downregulation in the case of a thin endometrium in patients with recurrent implantation failure.

## 2. Materials and Methods

### 2.1. Subjects

In accordance with the purpose of the study, two groups were formed from participants who applied to the Scientific Center of Obstetrics, Gynecology and Perinatology (Almaty, Kazakhstan) in 2023. An individual follow-up chart was compiled for each woman, including the physical examination findings; complaints; obstetric and gynecological history data; transvaginal ultrasound findings on the GEVOLUSONE (Vienna, Austria); laboratory test results, including the prothrombin index (PI) and time (PT), activated partial thromboplastin time (APTT), and international normalized ratio (INR); the hormone levels of luteinizing hormone (LH), follicle-stimulating hormone (FSH), thyroid-stimulating hormone (TSH), and prolactin; genetic counseling; and karyotyping.

The inclusion criterion for the RIF group was the presence of 3 or more implantation failures after an IVF procedure, as well as the presence of a thin endometrium (less than 7 mm during the implantation window during ultrasound examination). The fertile women had at least one term and healthy newborn and were without any reproductive losses in anamnesis.

### 2.2. Ethics Approval

This study was approved by the Ethical Committee of Al Farabi Kazakh National University, Kazakhstan (Code: IRBA400/IRB 00010790). All participants provided written informed consent for the use of biomaterials in this study.

### 2.3. Sample Processing

Endometrial samples were collected in the implantation window (LH+7 − LH+10) of the natural menstrual cycle from RIF patients with a thin endometrium (*n* = 20) and fertile women (*n* = 14). Endometrial tissue was collected using a Pipelle biopsy with a Goldstein catheter (SonoBiopsyTM J-GSBX-072026 size (Fr) 7.2 Cook Incorporated, Bloomington, IN, USA). Tissue samples were transferred from the catheter into a cryotube with 1 mL ribonucleic acid (RNA) later stabilization solution (Thermo Fisher Scientific, Vilnius, Lithuania) and stored in a refrigerator at 4 °C for 12 h. The next day, the samples were transferred to a freezer at −20 °C. After sample collection was completed, they were transferred to the M. Aitkhozhin Institute of Molecular Biology and Biochemistry (Almaty, Kazakhstan).

### 2.4. RNA Isolation from Endometrial Samples

The isolation of the polyA-mRNA fraction from ~5 mg of endometrial tissue was performed using the Dynabeads™ mRNA DIRECT™ Purification Kit (Thermo Fisher Scientific) according to the manufacturer’s “Mini” protocol and using liquid nitrogen to homogenize the sample. The amount of extracted mRNA ranged from 0.196 to 0.288 µg from sample to sample, which was consistent with the expected amounts. The isolated RNA was immediately subjected to further analysis.

### 2.5. Complementary Deoxyribonucleic Acid (cDNA) Synthesis and Quantitative Polymerase Chain Reaction (PCR)

cDNA was obtained using a High-Capacity cDNA Reverse Transcription Kit (Thermo Fisher Scientific), according to the manufacturer’s protocol. Quantitative PCR was performed in duplicate using ready-made primers and hydrolysis probes from the TaqMan™ Gene Expression Assay (Thermo Fisher Scientific, assay IDs: Hs00426339_m1, Hs01115665_m1, Hs00173634_m1, Hs00174103_m1, Hs02800695_m1, Hs00233987_m1, Hs00989291_m1, Hs00236937_m1, Hs01003716_m1, Hs00232769_m1, Hs02786624_g1, and Hs01122445_g1) and TaqMan Fast Advanced Master Mix (Thermo Fisher Scientific), under the conditions recommended by the manufacturer, on the StepOnePlus Real-Time PCR System (Thermo Fisher Scientific). The cycling parameters were as follows: 1 cycle at 50 °C for 2 min and 1 cycle at 95 °C for 20 s, followed by 40 cycles of denaturation at 95 °C for 1 s and annealing/extension at 60 °C for 20 s.

Primary processing was performed using the StepOnePlus 2.2.2 and ExpressionSuite v1.3 programs. The relative quantification of gene expression was carried out using the comparative Cq (ΔΔCq) method, with the modifications described by Königshoff M. et al. (2009) [25]. The relative transcript abundance was expressed in terms of ΔCq values (ΔCq = Cq_reference_ − Cq_target_). The *glyceraldehyde-3-phosphate dehydrogenase (GAPDH)* and *tyrosine 3-monooxygenase/tryptophan 5-monooxygenase activation protein zeta (YWHAZ)* housekeeping genes were used as a reference and internal controls, in accordance with a previous study [14]. The ΔΔCq value (ΔΔCq = ΔCq_case_ − ΔCq_control_) was considered as the log_2_ fold change.

In this study, we followed the Minimum Information for Publication of Quantitative Real-Time PCR Experiments (MIQE) guidelines’ recommendations [26].

### 2.6. Statistical Analysis

Most of the statistics were obtained with the Jamovi program v. 2.3.28 (https://www.jamovi.org, accessed on 5 September 2024). The statistical significance of the differences in ΔCq between the groups was calculated using the two-tailed Mann–Whitney U test. Spearman’s rank correlation method was used to examine the relationships between the quantitative variables, and *p* < 0.05 was considered statistically significant. For multiple comparisons, the online FDR correction calculator was used to adjust the *p*-values (https://www.sdmproject.com/utilities/?show=FDR, accessed on 5 September 2024). For the comparative visualization of the gene expression levels, box plots were constructed using a web tool, BoxPlotR (http://shiny.chemgrid.org/boxplotr, accessed on 5 September 2024). The characteristics of the markers were evaluated via ROC analysis using the web tool easy ROC v.1.3.1 [27] and Jamovi. The Youden’s index method was used to calculate the optimal cut-off points. The evaluation of the classifiers via the interpretation of the area under the ROC curve (AUC) was performed as described in [28].

## 3. Results

### 3.1. Clinical Data

Demographic and clinical characteristics of the study groups are presented in Table 1.

When the clinical characteristics were compared between the two groups, no statistical differences were found in terms of age and BMI. All participants were under 40 years of age. In the gynecological history, the presence of chronic salpingo-oophoritis and operations on the pelvic organs were more often noted in patients from the RIF group compared to the fertile group. There were no significant differences between the groups in terms of the hemostasis parameters or blood hormone levels.

### 3.2. The Level of mRNA in Endometrial Samples of RIF Patients in Comparison with the Fertile Group

The comparative gene expression statistics between the RIF patients and fertile women are presented in Table 2. The study found that the expression of 2 out of 10 genes differed significantly between the studied groups. It was found that the expression of *CXCL1* and *C4BPA* was significantly decreased in the RIF patients with a thin endometrium compared to the fertile women (log_2_ fold change = −1.95 and −2.09, *p* = 0.005 and 0.030, respectively) (Figure 1). However, after adjustment for multiple comparisons, only the difference in *CXCL1* expression remained significant (pFDR = 0.050 and 0.150, respectively). There were no significant differences in the expression of the *CXCL8*, *HPRT1*, *MMP10*, *INFG*, *VEGFB*, *HAND2*, *IL-15*, and *TNC* genes between the groups.

### 3.3. Associations with Clinical and Laboratory Characteristics

Next, we analyzed the differences in gene expression among groups with different clinical characteristics. Among the fertile women, the *C4BPA* gene expression was significantly increased and *HAND2* expression was significantly decreased in the endometrium in women with polyps compared to those without them (in both cases, *p* = 0.020). Among the RIF women, there were no significant differences in the gene expression levels in patients with and without endometriosis.

Spearman’s rank correlation method was used to examine the relationship between the expression of the studied genes and the quantitative laboratory characteristics in the two groups. The correlations are presented in Table 3. In the RIF patients with a thin endometrium, significant inverse moderate correlations were established between the *HAND2* and TSH levels (rho = −0.671, *p* = 0.001), *HAND2* and fibrinogen levels (rho = −0.465, *p* = 0.038), *VEGFB* and prolactin levels (rho = −0.446, *p* = 0.049), and *VEGFB* and TSH levels (rho = −0.526, *p* = 0.017). In the fertile group, there was an inverse moderate relationship between *HAND2* and the BMI (rho = −0.688, *p* = 0.008). It should be noted, however, that, after adjustment for multiple comparisons, no correlations remained significant.

### 3.4. Receiver Operating Characteristic (ROC) Analysis

We performed an ROC analysis to evaluate the potential of using the mRNA of genes differentially expressed in the endometria of RIF patients as markers for the prediction of RIF. The results are presented in Table 4. The areas under the ROC curve (AUCs) were obtained for *CXCL1* (0.782) and *C4BPA* (0.726) (Figure 2). The use of a combination of *C4BPA* and *CXCL1* resulted in an increase in the AUC to 0.806 (with specificity of 64.3% and sensitivity of 83.3%).

## 4. Discussion

Fertility impairments, including implantation failure, have been increasingly associated with a thin endometrium [29,30], which is characterized by the abnormal activation of the inflammatory milieu and a decreased oxidative stress response [31]. Therefore, it is of great significance to study the molecular mechanisms of signaling pathway dysfunction in patients with a thin endometrium.

We focused on 10 genes involved in endometrial receptivity immune pathways that showed altered regulation in the receptive phase of the endometrium in healthy women.

In this study, gene expression was examined in RIF patients with a thin endometrium together with the matched endometrial tissue of fertile women, and we found the abnormal activation of the inflammatory environment in those with a thin endometrium. It was found that the *C4BPA* and *CXCL1* genes’ expression was significantly decreased in RIF patients with a thin endometrium compared with fertile women (*p* = 0.030 and *p* = 0.005, respectively). The increased expression of the gene encoding *Complement Component 4 Binding Protein Alpha (C4BPA)* in healthy women leads to decreased activity of the complement system, which has a beneficial effect on embryo development [32]. Abnormally decreased levels of C4BPA were found in the mid-secretory endometria of women with implantation failure [18] and endometriosis [33]. In the thin endometrium, the abnormal activation of immune cells and an impaired cell dialogue with other cell types were accompanied by the increased activity of ligand–receptor pair genes in CD45 and complement signals [31]. It is also of interest to study the expression of chemokine family genes as a potential factor in endometrial thinning [31].

In our study, the downregulation of the *C-X-C motif chemokine ligand 1 (CXCL1)* gene indicated pro-inflammatory signaling cascade activation in RIF patients with a thin endometrium. At the same time, there were no significant differences in the concentrations of the *CXCL8*, *HPRT1*, *MMP10*, *INFG*, *VEGFB*, *HAND2*, *IL-15*, and *TNC* genes between the groups. Our findings indirectly indicate that a “silent endometrium” is observed in those with a thin endometrium. A thin endometrium and other pathological conditions affecting the uterine mucosa can lead to a refractory endometrium, in which there is no proliferation [34]. This situation is supported by a study of intercellular communication via CellChat, which showed that, in a thin endometrium, the number and strength of the interactions between stromal cells and proliferating stromal cells is reduced [31]. In the late proliferative phase, women with a thin endometrium and the reduced expression of PDZ-binding kinase in the endometrium were found to have suppressed proliferation processes [35].

In the current study, in RIF patients with a thin endometrium, significant inverse moderate correlations were shown. It was noted that a decreasing *HAND2* gene level was associated with increasing TSH and fibrinogen levels; in fertile women, the decrease in the *HAND2* gene level was associated with an increase in BMI.

*Heart and neural crest derivative-expressed transcript 2 (HAND2)* has a regulatory effect on interleukin 15 (IL-15), which in turn plays a direct role in the activation and survival of uterine natural killer cells. The levels of HAND2 and IL-15 are significantly increased in the secretory phase in healthy women [23]. The same trends were observed in *VEGFB* with the TSH and prolactin levels. VEGF-B has been shown to be a survival factor for vascular endothelial cells [21]. These findings confirm that the RIF population is extremely heterogeneous and includes various risk factors associated with lifestyle factors, overweight, thrombophilia, and pathological conditions of the reproductive system (endometriosis, polyps, adenomyosis, and polycystic ovary disease) [36,37]. Future studies on endometrial receptivity should be performed while taking these factors into account. A receptive endometrium is characterized by the presence of a complex interactive network at different stages of the implantation process, which is reflected in the high variability of the phenotype [38]. However, our study highlights the prognostic potential of the *C4BPA + CXCL1* gene combination in predicting RIF.

Our study had some limitations that need to be addressed in future studies. It used a small sample size, and it is recommended to increase the observational cohort according to the clinical form. The transcriptional analysis of thin endometrial tissue should be validated using multi-omics data, including proteomics (cell immunophenotyping) or metabolomics.

## 5. Conclusions

Our study revealed the differential expression of the *C4BPA* and *CXCL1* genes in RIF patients with a thin endometrium compared with the endometria of fertile women. These data allowed us to evaluate the predictive value of the combined model in predicting implantation failure depending on the endometrial thickness. The data also provide potential therapeutic targets for the treatment of a thin endometrium in patients undergoing assisted reproductive technology programs.

## Figures and Tables

**Figure 1 diagnostics-14-01967-f001:**
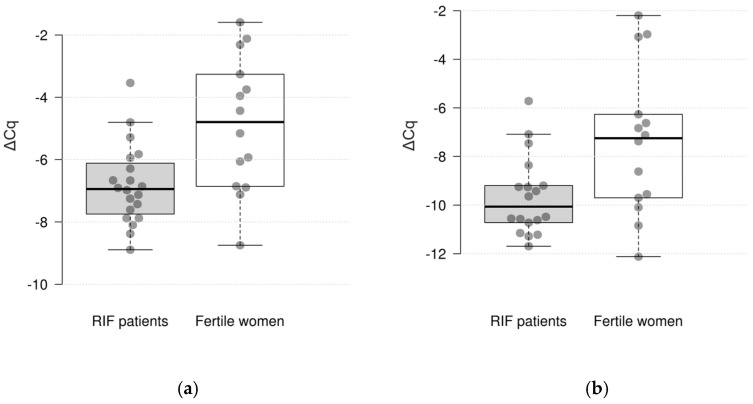
Differences in gene expression levels (ΔCq values) in endometrial tissue between RIF patients and fertile women: (**a**) *CXCL1* gene; (**b**) *C4BPA* gene.

**Figure 2 diagnostics-14-01967-f002:**
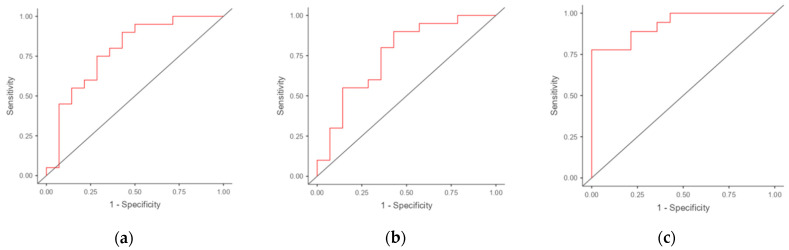
ROC curves for potential markers and combinations of markers in discriminating RIF patients from fertile women: (**a**) *CXCL1*; (**b**) *C4BPA*; (**c**) combination of *CXCL1* and *C4BPA*.

**Table 1 diagnostics-14-01967-t001:** Demographic and clinical characteristics of the study groups.

Characteristics	RIF Patients (*N* = 20)	Fertile Women (*N* = 14)
Age (years): Mean ± SD	34.65 ± 5.29	32.71 ± 5.47
BMI (kg/m^2^): Mean ± SD	23.36 ± 4.24	23.98 ± 3.81
Endometrium thickness (mm): Mean ± SD	6.51 ± 1.27	7.74 ± 0.88
Fibrinogen, g/L: Mean ± SD	3.05 ± 0.51	2.71 ± 0.17
Protrombin index, %: Mean ± SD	99.16 ± 8.22	97.62 ± 7.81
Protrombin time, s: Mean ± SD	13.04 ± 2.35	12.47 ± 3.27
Activated partial thromboplastin time, s: Mean ± SD	34.04 ± 3.61	25.15 ± 13.87
International normalized ratio: Mean ± SD	1.04 ± 0.12	1.06 ± 0.03
LH, IU/L: Mean ± SD (normal level 0.61–16.3)	8.34 ± 2.13	-
FSH, IU/L: Mean ± SD (normal level 4.7–21.5)	6.19 ± 1.43	-
Prolactin, ng/mL: Mean ± SD (normal level—less than 25)	325.0 ± 152.4	-
TSH, mIU/L: Mean ± SD (normal level 0.4–4.0)	2.84 ± 0.74	-
Chronic endometritis, yes/no	20/0	0/14
Chronic salpingo-oophoritis, yes/no	10/10	0/14
Pelvic organs surgeries, yes/no	10/10	0/14
Ectopic pregnancy, yes/no	2/18	0/14
Endometriosis, yes/no	5/15	1/13
Uterine fibroids, yes/no	1/19	0/14
Polyps, yes/no	1/19	6/8

Note. BMI—body mass index; LH—luteinizing hormone; FSH—follicle-stimulating hormone; TSH —thyroid-stimulating hormone.

**Table 2 diagnostics-14-01967-t002:** Comparative gene expression statistics between RIF and fertile women.

Gene	Cq Mean ± SD		ΔCq Mean ± SE		ΔΔCq (95% CI) log_2_ Fold Change	*p*-Value
	RIF	Fertile	RIF	Fertile		
*CXCL1*	27.84 ± 4.27	27.76 ± 3.84	−6.82 ± 0.29	−4.87 ± 0.58	**−1.95 (−3.45; −0.60)**	0.005 **
*C4BPA*	29.69 ± 3.29	30.28 ± 4.21	−9.65 ± 0.38	−7.39 ± 0.81	**−2.09 (−3.99; −0.38)**	0.030 *
*IL15*	25.89 ± 3.89	26.86 ± 2.94	−4.87 ± 0.22	−3.97 ± 0.32	−0.73 (−1.65; 0.00)	0.051
*IL8*	28.16 ± 4.00	28.60 ± 4.44	−7.55 ± 0.54	−5.71 ± 1.00	−1.67 (−3.47; 0.18)	0.060
*TNC*	25.07 ± 4.49	28.20 ± 4.21	−4.04 ± 0.40	−5.30 ± 0.60	1.48 (−0.45; 3.27)	0.112
*VEGFB*	24.20 ± 3.41	25.60 ± 2.50	−3.18 ± 0.21	−2.71 ± 0.27	−0.45 (−1.06; 0.18)	0.148
*IFNG*	33.58 ± 2.85	34.86 ± 3.11	−13.52 ± 0.36	−12.51 ± 0.52	−0.89 (−2.17; 0.30)	0.157
*HPRT1*	24.77 ± 4.23	26.31 ± 3.29	−3.75 ± 0.13	−3.42 ± 0.33	−0.32 (−1.13; 0.12)	0.180
*HAND2*	22.90 ± 3.52	24.48 ± 2.73	−1.88 ± 0.17	−1.59 ± 0.31	−0.31 (−0.87; 0.23)	0.231
*MMP10*	29.76 ± 5.55	31.22 ± 4.82	−8.74 ± 0.66	−8.33 ± 1.11	−0.63 (−2.90; 1.95)	0.904

Note. * significant *p* < 0.05; ** significant *p* after FDR correction for multiple comparisons.

**Table 3 diagnostics-14-01967-t003:** Detected correlations between gene expression levels and clinical and laboratory parameters.

Group	Parameter 1	Parameter 2	Spearman Rho	*p*-Value
RIF group	*VEGFB*	Prolactin	−0.446	0.049
	*VEGFB*	Thyroid-stimulating hormone	−0.526	0.017
RIF group	*VEGFB*	Prolactin	−0.446	0.049
	*VEGFB*	Thyroid-stimulating hormone	−0.526	0.017
	*HAND2*	Thyroid-stimulating hormone	−0.671	0.001
	*HAND2*	Fibrinogen	−0.465	0.038
Fertile group	*HAND2*	Body mass index	−0.688	0.008

**Table 4 diagnostics-14-01967-t004:** ROC analysis results.

Potential Markers and Combinations	AUC (95% CI)	Optimal Cut-Off Point	Sensitivity (95% CI)	Specificity (95% CI)
*CXCL1*	0.782 (0.613–0.951)	−5.286	0.900 (0.683–0.988)	0.571 (0.289–0.823)
*C4BPA*	0.726 (0.535–0.918)	−7.462	0.889 (0.653–0.986)	0.571 (0.289–0.823)
*C4BPA + CXCL1*	0.806	-	0.833	0.643

## Data Availability

The data presented in this study are available on request from the corresponding author due to privacy.

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
