# Peer review of "Altered Expression of C4BPA and CXCL1 Genes in the Endometrium of Patients with Recurrent Implantation Failure after In Vitro Fertilization and Thin Endometrium"

_diagnostics, 2024, doi:10.3390/diagnostics14171967_

Round 1

Reviewer 1 Report

Comments and Suggestions for Authors

1. The number of patients on both groups; the study and the control one are small generalized the results.

2. In table 1, I do prefer to write the normal values of the hormones mentioned

Reviewer 2 Report

Comments and Suggestions for Authors

Overall, the work's idea is interesting, and it could impact related fields. However, several concerns should be addressed.

 General comments:

1. The authors should ensure that all gene names are italicized throughout the manuscript, including the Title, to match the standards of HUGO for gene nomenclatures.

2. The authors should ensure that all acronyms, particularly the gene names, are spelled out on their first mention in the text.

3. The authors are advised to copyedit the manuscript to meet a standard of professional English: a native English-speaking colleague could help them with this issue, if possible, or they may need to use a professional language editing service.

Introduction

This section needs extensive revision, as it lacks coherence. The flow of the idea was not logical, and it was hard to follow the authors' elaborations.

-       Lines 55 and 56:” differentially expressed genes varied from 107 to 2878,…”What did these figures indicate? Please clarify, as not all future readers will be specialty-related ones.

-       Line 60: “bioinformatics prediction” is a vague expression. What are the names of the tools (databases) applied to identify the mentioned genes and their confidence levels?

-       Line 63: “ER Map®/ER Grade®” What is the vendor of this panel? Please clarify with the official citation in the text (Company name, City, Country).

-       Line 72: “In this study, we focused on 10 genes also identified.” The reason behind the selection of these 10 genes was not clear. Also, the function and role of each one in this context were unclear.

 Methods

-       Ther detailed characteristics of the study groups (Table 1) are better presented in the “Results section.”

-       Table footer should be provided to clarify the type of data presentation and the expansion of any abbreviations mentioned in the table. (e.g. BMI).

Lines 103 and 104: “lab tests, as well as diagnostic tests where indicated: pelvic ultrasound, blood hormone level testing” should be expanded and mentioned in detail as part of data transparency.

-       Lines 114-118: should be in the “Results section.”

-       Did the authors measure the concentration and quality of the extracted RNA before proceeding to RT-PCR? If so, please specify the range of extracted RNA measured and its integrity check.

The PCR components and concentrations should be detailed as part of the data transparency to facilitate work replication by interested readers. Also, the PCR programming should be detailed.

It was not clear whether the applied primer sequences were designed by the authors or purchased as ready forms. Please provide the primer sequences for the 10 genes as a main Table.

-       The authors should confirm they apply all the quality control measurements required for the qPCR according to MIQE guidelines.

-       According to the standard nomenclatures related to qPCR, the “Ct” = threshold cycle terminology should be replaced by “Cq” = quantitative cycle.

-       Which type of internal controls did the authors apply during the PCR runs?  

Discussion

 Based on the iThenticate report, this section contains a high percentage of similar text. The authors should revise and write this section in their own words.

Minor comments

-       Please revise the measuring unit of BMI (kg/m2) to be (kg/m2)

-       Line 258: delete (p< 0.05)

Comments on the Quality of English Language

The authors are advised to copyedit the manuscript to meet a standard of professional English: a native English-speaking colleague could help them with this issue, if possible, or they may need to use a professional language editing service.

Reviewer 3 Report

Comments and Suggestions for Authors

The idea of the study is clear and really important from the clinical perspective. References are properly choosen and up-to-date.

There are rather small mistakes which can be easily fixed prior final decision concerning publication. In my opinion paper should be recommended for publication after introduction small corrections.

Professional language editor should revise paper as there are some mistakes, sloppy usage of words or technical/editorial issues. Especially, but not only in the abstract part (wrong sentences construction, usage of coma instead of dots, etc.)

Missing company details, while they should be given in full names, more precise.

Explanation and/or clarification of asterisks usage in the figures/tables. What abbreviations/acronyms stand for. If the Authors would like, they may point out that explanations are given in particular parts of the manuscript...

Significant results presented in the figures may be presented in bold font.

What for to present GAPDH and YWHAZ in the table?

Comments on the Quality of English Language

Professional language editor should revise paper as there are some mistakes, sloppy usage of words or technical/editorial issues. Especially, but not only in the abstract part (wrong sentences construction, usage of coma instead of dots, etc.)

Round 2

Reviewer 2 Report

Comments and Suggestions for Authors

The manuscript showed improvement. Thanks to the authors for addressing the raised concerns. In the proof stage, please italicize the names of the genes in the main Title.